# DuetServe: Harmonizing Prefill and Decode for LLM Serving via Adaptive GPU Multiplexing

Lei Gao [1]  Chaoyi Jiang [1]  Hossein Entezari Zarch [1]  Daniel Wong [2]  Mark D. Hill [3]  Murali Annavaram [1]

## Abstract

Modern LLM serving systems must sustain high throughput while meeting strict latency SLOs across two distinct inference phases: compute-intensive prefill and memory-bound decode phases. Existing approaches either (1) aggregate both phases on shared GPUs, leading to interference between prefill and decode phases, which degrades Time-Between-Tokens (TBT); or (2) disaggregate the two phases across GPUs, improving latency but wasting resources through duplicated models and KV cache transfers. We present DuetServe, a unified LLM serving framework that achieves disaggregation-level isolation within a single GPU. DuetServe operates in aggregated mode by default and dynamically activates SM-level GPU spatial multiplexing when TBT degradation is predicted. Its key idea is to decouple prefill and decode execution only when needed through fine-grained, adaptive SM partitioning that provides phase isolation only when contention threatens latency service level objectives. DuetServe integrates (1) an attention-aware roofline model to forecast iteration latency, (2) a partitioning optimizer that selects the optimal SM split to maximize throughput under TBT constraints, and (3) an interruption-free execution engine that eliminates CPU–GPU synchronization overhead. Evaluations show that DuetServe improves total throughput by up to $1.3\times$ while maintaining low generation latency compared to state-of-the-art frameworks.

[1]University of Southern California [2]University of California, Riverside [3]University of Wisconsin–Madison. Correspondence to: Murali Annavaram <annavara@usc.edu>.

*Proceedings of the 43rd International Conference on Machine Learning*, Seoul, South Korea. PMLR 306, 2026. Copyright 2026 by the author(s).

## 1. Introduction

Large Language Models (LLMs) have unlocked unprecedented capabilities, powering applications such as conversational assistants (OpenAI et al., 2024a; Gemini-Team et al., 2024), creative content generation (OpenAI et al., 2024b; Liu et al., 2024), and autonomous agents (Li et al., 2023; Luo et al., 2025). User requests in these systems are served in real time by large GPU clusters in the cloud. As these systems scale to millions of concurrent sessions, ensuring efficient use of GPU resources while meeting strict Service Level Objectives (SLOs) for real-time responses becomes increasingly difficult—fast response times typically require over-provisioning system resources.

LLM inference is commonly divided into two phases with distinct resource profiles. The *prefill* phase encodes the input prompt in a single forward pass and is typically compute-bound, benefiting from large batches that saturate GPU streaming multiprocessors (SMs). The *decode* phase generates output tokens iteratively; each step repeatedly reads and writes the key–value (KV) cache, and thus decoding is often constrained by high-bandwidth memory (HBM) capacity and bandwidth. This phase asymmetry directly shapes two standard serving metrics: Time-to-First-Token (TTFT), largely determined by prefill, and Time-Between-Tokens (TBT), determined by decode. To improve request throughput and GPU utilization, modern frameworks adopt iteration-level scheduling such as *continuous batching* (Yu et al., 2022), which admits and removes requests at each decoding step so that requests with shorter outputs can complete without waiting for longer ones. However, inserting long prefills into an ongoing decode batch synchronously couples the two phases, which can severely inflate TBT and lead to SLO violations.

A prominent mitigation is *chunked prefill* (Agrawal et al., 2024b), which bounds per-iteration work using a token budget and schedules long prefills in smaller chunks. The token budget is commonly chosen to maximize linear-layer utilization (often at the "knee" of a roofline curve) and has increased substantially on newer GPUs (e.g., from 2K tokens on A100 to 8K tokens on H100). Yet, fully consuming a large budget can still yield long iteration latency, directly inflating decode TBT under synchronous execution, and the

linear-layer-dominance assumption becomes increasingly inaccurate for long-context requests where attention and KV-cache access dominate latency (Section 3). As an alternative, *prefill–decode disaggregation* isolates the two phases on separate GPU clusters (Zhong et al., 2024; Patel et al., 2024; Qin et al., 2025), but it introduces KV-transfer and memory duplication overheads and can suffer from resource imbalance under time-varying traffic (e.g., excess prefill capacity under decode-heavy workloads, or the reverse). These limitations motivate an adaptive design that achieves disaggregation-like isolation only when needed, without permanently incurring disaggregation overhead.

We introduce **DuetServe**, an adaptive serving framework that operates in an aggregated mode by default and dynamically enables GPU spatial multiplexing when an SLO violation, particularly a TBT degradation, is predicted. The key idea of DuetServe is to decouple prefill and decode execution *within a single GPU* through fine-grained SM-level partitioning, providing phase isolation only when contention threatens latency SLOs. Leveraging modern GPU capabilities, the system logically partitions hardware resources within a single device by dividing GPU SMs into distinct, concurrently executing sets. These logical "sub-GPUs" isolate prefill and decode workloads during periods of interference and revert to fully shared execution when contention subsides, maintaining both high throughput and low latency.

Our contributions are as follows. DuetServe integrates three tightly coupled components, which we validate through a comprehensive evaluation: (1) an attention-aware roofline analytical model that predicts iteration latency based on operator-level compute and memory characteristics, allowing the scheduler to detect potential TBT violations in advance; (2) a dynamic GPU partitioning optimizer that determines the optimal allocation of SMs between prefill and decode to balance utilization and latency, activating GPU spatial multiplexing only when necessary; and (3) an interruption-free execution engine that employs look-ahead decode scheduling and asynchronous kernel dispatch to eliminate CPU–GPU synchronization overhead and ensure uninterrupted concurrent execution. (4) We evaluate DuetServe on Qwen3-8B and Qwen3-14B using three realistic workloads and show up to $1.3\times$ total throughput improvement while maintaining low TBT latency compared to the state-of-the-art LLM serving frameworks.

## 2. Background

**LLM Inference Process.** Let $x \in \mathbb{R}^{s \times d}$ denote a single input request, where $s$ is the sequence length (i.e., the number of prompt tokens) and $d$ is the embedding dimension. The forward pass of a Transformer block in typical LLMs such as Qwen (Yang et al., 2025) and Llama (Grattafiori

et al., 2024) can be expressed as:

$$
\begin{aligned}
Q = xW_q, \ K = xW_k, \ V = xW_v, & \quad W_{q,k,v} \in \mathbb{R}^{d \times d}, \\
u = \text{Softmax}\left(QK^\top/\sqrt{d}\right)VW_o, & \quad W_o \in \mathbb{R}^{d \times d}, \\
v = \text{LayerNorm}(u + x; \gamma_1, \beta_1), & \quad \gamma_1, \beta_1 \in \mathbb{R}^d, \\
z = \sigma(vW_1)W_2, & \quad W_1, W_2^T \in \mathbb{R}^{d \times m}, \\
y = \text{LayerNorm}(z + u + x; \gamma_2, \beta_2), & \quad \gamma_2, \beta_2 \in \mathbb{R}^d.
\end{aligned}
$$

During the prefill phase, the generated $K$ and $V$ matrices are stored in the KV cache. In the decode phase, the decoder layer processes one token at a time, with the input $x \in \mathbb{R}^{1 \times d}$. The KV cache is updated by concatenating the newly computed key and value pairs with the existing ones before the attention operation. The rest of the operations in the decoding phase are identical to those in the prefill phase.

**GPU Spatial Multiplexing.** Modern GPUs expose massive parallelism through hundreds of SMs, which can be logically partitioned to support multi-tenant execution. Several mechanisms have been developed to enable such partitioning. Multi-Instance GPU (MIG) (NVIDIA, 2025e) statically divides a GPU into multiple isolated instances, each with dedicated compute and memory resources. Multi-Process Service (MPS) (NVIDIA, 2025f) supports coarse-grained spatial sharing by assigning fixed groups of SMs to different processes. However, these mechanisms cannot be applied within a single process, and the overhead of reconfiguring partitions is high, making them unsuitable for dynamic and heterogeneous workload patterns. GreenContext (NVIDIA, 2025d) introduces intra-process SM partitioning by allowing different kernels or streams within the same GPU context to execute concurrently on disjoint subsets of SMs. While it can be used for on-demand spatial multiplexing, fast reconfiguration requires pre-creating multiple contexts, leading to additional memory overhead and reduced flexibility.

In contrast, our approach achieves the same goal with minimal runtime overhead by directly controlling the number of SMs available to each kernel or stream at launch time. This is enabled by libsmctrl (Bakita & Anderson, 2023), a recently developed low-level library that allows selectively masking SMs visible to individual kernels or streams. Operating at the driver level, libsmctrl provides transparent and fine-grained control over GPU resource allocation without requiring any modification to the application or model code.

## 3. Motivation

**Chunked Prefill Limitations.** Sarathi-Serve (chunked prefill) (Agrawal et al., 2024b) assumes that linear layers dominate inference latency, and adopts a token-budget scheduler to bound per-iteration work. At each iteration, a fixed token budget is set. The scheduler first schedules as

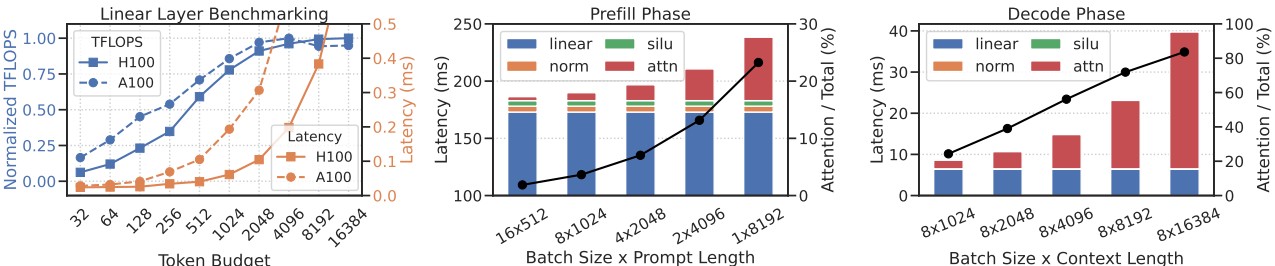

*Figure 1.* (a) Linear layer benchmarking shows A100 and H100 saturate near 2K and 8K tokens. (b) Prefill latency under an 8K budget violates TBT SLOs despite full utilization. (c) Decode latency rises with longer contexts as KV cache grows even under the same budget.

many decode requests as possible, where each scheduled decode request consumes one token from the budget. If any token budget remains, prefill requests are scheduled, and each prefill request consumes a number of tokens equal to its prompt length. If the remaining token budget is smaller than the prompt length, the prefill request is chunked to exactly fill the remaining budget. Because linear layers are agnostic to whether tokens originate from prefill or decode, the scheduler treats the total number of tokens as the primary determinant of compute load. The value of the token budget is chosen to maximize linear-layer utilization, typically at the "knee" of a roofline curve obtained by profiling a linear layer on the target GPU.

Figure 1(a) illustrates that the saturation point of a $4096 \times 4096$ linear layer shifts from around $T = 2048$ tokens on A100 to around $T = 8192$ tokens on H100, motivating larger default budgets (e.g., vLLM uses 2048 on A100 and 8192 on H100). However, using a budget of 8192 can still produce long per-iteration latency for end-to-end model execution: in Figure 1(b), prefill-only batches under the same 8192-token budget consistently exceed 180 ms. When such a batch is mixed with decode, the decode TBT is inflated to a similar level, violating a typical 100 ms TBT SLO (Agrawal et al., 2024b). Meeting the TBT constraint thus requires reducing the token budget, which improves latency but lowers linear-layer utilization and compromises throughput efficiency.

***Observation 1:*** *Fully consuming the token budget can violate TBT SLOs despite full linear layer utilization, while reducing the token budget improves latency at the cost of throughput efficiency.*

The second trend is that the linear layer may not always dominate model latency. Since the prefill attention operation cost grows quadratically with prompt length, the attention cost cannot be controlled under a fixed token budget. As shown in Figure 1(b), although all settings use the same token budget during prefill, in the last case with a single 8192-token prefill, the attention module accounts for approximately 25% of the total forward latency. The problem becomes more severe during the decode phase. Figure 1(c)

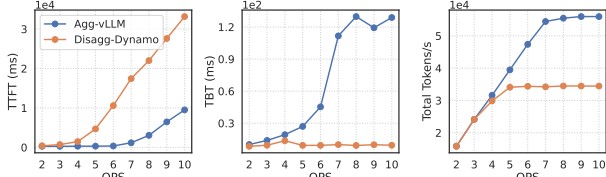

*Figure 2.* Performance comparison between PD aggregated and disaggregated systems across varying QPS.

shows profiling latency for decode-only batch execution. Although all settings use the same token budget of 8, they exhibit more than $4\times$ latency variation as the context length increases. This occurs because attention latency scales with KV cache size, and KV cache reads dominate runtime as the context grows. As a result, a token-budget-based scheduler, which is derived to maximize the linear layer utilization, is sub-optimal as context length increases.

***Observation 2:*** *The token-budget-based scheduling policy ignores attention cost during both prefill and decode phases, failing to provide consistent TBT latency control.*

**PD Disaggregation Limitations.** We benchmark both prefill-decode (PD) aggregated with chunked prefill (Agg-vLLM) and PD disaggregated (Disagg-Dynamo) systems using Qwen3-8B on two H100 GPUs (vLLM, 2025b; NVIDIA, 2025c). In the aggregated setup, both GPUs host identical model replicas under round-robin request dispatch with a token budget of 8192. In the disaggregated setup, we adopt a 1P+1D configuration, where one GPU handles all prefill operations and the other handles decode, fully isolating the two phases. The workload consists of 8000 input tokens and 200 output tokens per request, following the official vLLM disaggregation demo (LMCache, 2025). We vary the query-per-second (QPS) rate and measure TTFT, TBT, and total token throughput.

As shown in Figure 2, although TBT remains relatively stable in the disaggregated system as QPS increases, TTFT rises sharply when QPS > 4, whereas the aggregated system starts to saturate at QPS = 7. This trend aligns with

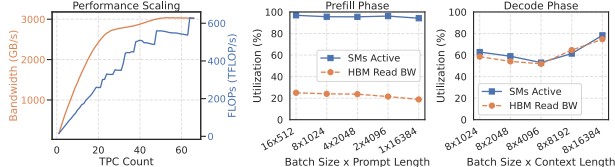

*Figure 3.* (a) Profiled HBM bandwidth and FLOPs versus active TPCs. (b–c) Resource utilization during prefill and decode phases.

the widening throughput gap, where `Disagg-Dynamo` achieves less than half the total tokens per second of `Agg-vLLM`. The root cause is resource imbalance: in the colocated setup, both GPUs execute prefill operations concurrently, effectively doubling prefill throughput. In contrast, the disaggregated setup dedicates only one GPU to prefill, making it the system bottleneck. As QPS increases, the prefill worker's throughput lags behind that of the decode worker, resulting in overall system throughput degradation.

***Observation 3:*** *PD disaggregated systems satisfy the TBT SLO but underutilize GPUs, reducing overall request throughput.*

**Opportunity.**  Figure 3(a) shows the scaling of effective HBM bandwidth and FLOPs as the number of active Texture Processor Clusters (TPCs) increases on H100. Each TPC contains two SMs and serves as the smallest unit for SM partitioning. We measure the achieved throughput using `cudaMemcpy` and `gemm` microbenchmarks. The HBM bandwidth utilization rises super-linearly with the number of active SMs; for instance, 20% of SMs already achieve about 60% of peak bandwidth. In contrast, FLOPs scale roughly linearly with SM count, except for minor quantization effects from discrete TPC activation granularity.

Figures 3(b) and (c) compare end-to-end GPU utilization during prefill and decode. The compute-bound prefill phase saturates SM resources but leaves most HBM capacity underutilized. Conversely, the memory-bound decode phase exhibits the opposite trend: high HBM traffic with under-occupied SMs. ***This complementary behavior reveals an opportunity for resource co-execution, where compute-intensive and memory-intensive workloads can share SM and HBM resources more efficiently.***

## 4. Method

Figure 4 presents an overview of DuetServe. At each iteration, the DuetServe scheduler begins with conventional chunked prefill scheduling. It first prioritizes ongoing decode requests, rescheduling them before admitting waiting prefill requests to fill the remaining token budget. If the remaining budget cannot accommodate an entire prompt, the scheduler automatically chunks the request and performs a partial prefill.

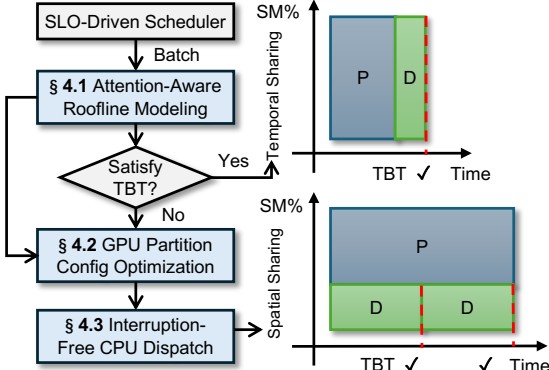

*Figure 4.* Overview of DuetServe.

After this step, the scheduler evaluates whether the current batch risks violating the decode TBT SLO. Using the attention-aware roofline model (Section 4.1), it predicts the expected latency of the prefill-decode mixed batch. If the predicted latency exceeds the TBT bound, DuetServe proactively activates GPU spatial multiplexing (Section 4.2). The workload is then divided into two disjoint sets, a decode-only batch and a prefill-only batch, each assigned to a separate, non-overlapping SM partition for concurrent execution (Section 4.3). The core idea is to decouple decode from prefill computation, allowing decode kernels to progress independently without being blocked or synchronized at every Transformer layer.

### 4.1. Attention-Aware Roofline Analytical Modeling

To predict potential TBT violations and guide SM partitioning, DuetServe employs an attention-aware roofline model that estimates model forward latency based on operator, compute, and memory characteristics. Operators are categorized as token-level, sequence-level, or communication operators (Agrawal et al., 2024a).

**Token-Level Operators.**  Token-level operators depend only on the total number of processed tokens (prefill plus decode) in the batch. These include linear projections, layer normalization, and activation functions. For the linear operator, given $n$ total tokens, embedding dimension $d$, linear input dimension $d_i$, output dimension $d_o$, and element size $b$, the operation and memory costs are

$$F_{lin} = 2nd_id_o, \qquad B_{lin} = nd_ib + d_id_ob + nd_ob$$

The linear operator accounts for input, output, and full weight tensor movements. This formulation applies to QKV projection ($d_i = d_o = d$), output projection ($d_i = d_o = d$), gate-up projection ($d_i = d, d_o = m$), and down projection ($d_i = m, d_o = d$). Other token-level operators, such as layer normalization and activation functions, follow the same modeling approach with operator-specific FLOP and

memory expressions. The latency of token-level operators in general is estimated as $t_{\text{tok}} = \max(F_{\text{tok}}/\Pi_{\text{SM}}, B_{\text{tok}}/\mathcal{B}_{\text{HBM}})$, where $\Pi_{\text{SM}}$ denotes the compute throughput of active SMs, and $\mathcal{B}_{\text{HBM}}$ denotes the achievable HBM bandwidth of the active SMs.

**Sequence-Level Operators.** The attention operation depends on both query and key-value sequence lengths of each request within each batch. For number of attention heads $h_q$, key-value heads $h_{kv}$, and head dimension $d_h = d/h_q$, the per-request FLOPs and memory bytes are given by:

$$F_{\text{attn/req}}(q, c) = 4h_q q(q+c)d_h + 2h_q q(q+c),$$
$$B_{\text{attn/req}}(q, c) = 2h_q q d_h b + 2h_{kv}(q+c)d_h b,$$

where $q$ denotes the number of scheduled query tokens and $c$ is the number of cached key-value tokens. The first term in $F_{\text{attn/req}}$ accounts for the dominant matrix multiplication operations in query-key similarity computation and attention-value aggregation, while the second term represents element-wise softmax and scaling operations. The memory term $B_{\text{attn/req}}$ includes reads and writes for query, key, value, and output tensors.

The estimator iterates over the batch, and applies the roofline model to compute attention latency of each request and aggregate the latency:

$$t_{\text{attn}} = \sum_{r=1}^{|\mathcal{R}|} \max\left( \frac{F_{\text{attn/req}}(q_r, c_r)}{\Pi_{\text{SM}}}, \frac{B_{\text{attn/req}}(q_r, c_r)}{\mathcal{B}_{\text{HBM}}} \right),$$

where $|\mathcal{R}|$ is the number of scheduled requests, and $(q_r, c_r)$ represent the query and cached token lengths of request $r$.

This modeling is flexible and captures prefill ($q > 1, c = 0$), chunked prefill ($q > 1, c > 0$), and decode ($q = 1, c > 0$) attention operations for different types of requests.

**Communication Operators.** When scaling LLM serving across multiple GPUs within a single node, tensor parallelism is widely adopted due to its high efficiency. It requires inter-GPU communication for synchronizing the attention linear layer output $u$ and FFN output $z$ (Section 2). We model this communication cost using a ring AllReduce latency formulation:

$$t_{\text{allreduce}} = 2(N-1)\alpha + \frac{2(N-1)B_{\text{lin\_o}}}{N\mathcal{B}_{\text{NVLink}}} + \frac{N(N-1)B_{\text{lin\_o}}}{\Pi_{\text{SM}}},$$

where $N$ is the number of GPUs, $\alpha$ is the startup latency profiled for the specific GPU platform (e.g., 3 $\mu$s for H100), $B_{\text{lin\_o}}$ is the linear operator's output tensor size in bytes, and $\mathcal{B}_{\text{NVLink}}$ denotes the aggregate unidirectional bandwidth of all NVLink connections on a GPU.

The first term captures the fixed startup cost of initiating communication, including link delay and synchronization

---

**Algorithm 1** DuetServe Scheduling Algorithm

1: **Input:** Scheduled prefill and decode requests $\mathcal{R}_{\text{mixed}}$, TBT SLO $\tau_{\text{TBT}}$, total SM count $S$
2: $t_{\text{mixed}}(S) = f_{\text{roofline}}(\mathcal{R}_{\text{mixed}}, \Pi_{\text{SM}}(S), \mathcal{B}_{\text{HBM}}(S))$
3: **if** $t_{\text{mixed}}(S) \leq \tau_{\text{TBT}}$ **then**
4:     GPU_temporal_sharing_execute($\mathcal{R}_{\text{mixed}}$, $S$)
5: **else**
6:     $\mathcal{R}_{\text{prefill}}, \mathcal{R}_{\text{decode}} \leftarrow \mathcal{R}_{\text{mixed}}$
7:     $\rho^\star \leftarrow 0, \mathcal{C}^\star \leftarrow \emptyset$
8:     **for** $S_d$ in range(2, $S + 1$, 2) **do**
9:         $t_d(S_d) = f_{\text{roofline}}(\mathcal{R}_{\text{decode}}, \Pi_{\text{SM}}(S_d), \mathcal{B}_{\text{HBM}}(S_d))$
10:         **if** $t_d(S_d) > \tau_{\text{TBT}}$ **then**
11:             continue
12:         **end if**
13:         $S_p \leftarrow S - S_d$
14:         $t_p(S_p) = f_{\text{roofline}}(\mathcal{R}_{\text{prefill}}, \Pi_{\text{SM}}(S_p), \mathcal{B}_{\text{HBM}}(S_p))$
15:         **for** $k$ in $\left( \lfloor \frac{t_p(S_p)}{t_d(S_d)} \rfloor, \lfloor \frac{t_p(S_p)}{t_d(S_d)} \rfloor + 1 \right)$ **do**
16:             $\rho \leftarrow \dfrac{k\, T_{\text{decode}} + T_{\text{prefill}}}{\max(k\, t_d(S_d), t_p(S_p))}$
17:             **if** $\rho > \rho^\star$ **then**
18:                 $\rho^\star \leftarrow \rho, \mathcal{C}^\star \leftarrow (S_p, S_d, k)$
19:             **end if**
20:         **end for**
21:     **end for**
22:     GPU_spatial_sharing_execute($\mathcal{R}_{\text{prefill}}, \mathcal{R}_{\text{decode}}, \mathcal{C}^\star$)
23: **end if**

---

overheads; the second term models the data transfer time constrained by NVLink bandwidth; and the third term accounts for the local reduction operations performed on each GPU. This formulation reflects the $2(N-1)$ communication rounds in a ring AllReduce protocol, consisting of a scatter-reduce phase followed by an all-gather phase, where each GPU exchanges partial results of size $B_{\text{lin\_o}}/N$ with its two neighbors.

**Overall Estimation.** DuetServe estimates Transformer block latency by summing the compute and communication costs of all sub-operations. The total model latency is then approximated as: $t_{\text{total}} = L \cdot t_{\text{block}} + t_{\text{cls}}$, where $L$ is the number of layers and $t_{\text{cls}}$ is the latency of the final linear classifier, modeled as a linear operator with input dimension $d_i = d$ and output dimension $d_o = \text{vocab\_size}$.

### 4.2. GPU Partitioning Configuration Optimization

Once the roofline model predicts a TBT violation, DuetServe determines how to divide GPU resources between prefill and decode to maintain latency guarantees while maximizing overall throughput.

At initialization, DuetServe profiles the achievable compute throughput $\Pi_{\text{SM}}(S)$ and memory bandwidth $\mathcal{B}_{\text{HBM}}(S)$ for

each possible SM partition size $S$. For a candidate split assigning $S_d$ SMs to decode and $S_p = S - S_d$ SMs to prefill, the predicted latencies follow the roofline model:

$$t_p(S_p) = f_{\text{roofline}}\big(\mathcal{R}_{\text{prefill}}, \Pi_{\text{SM}}(S_p), \mathcal{B}_{\text{HBM}}(S_p)\big),$$
$$t_d(S_d) = f_{\text{roofline}}\big(\mathcal{R}_{\text{decode}}, \Pi_{\text{SM}}(S_d), \mathcal{B}_{\text{HBM}}(S_d)\big).$$

When proactive SM partitioning is triggered, the decode TBT SLO is already at risk of violation, and the system typically operates in the regime where $t_p(S_p) > t_d(S_d)$. To mitigate idle periods and balance utilization, we execute $k$ decode steps on $S_d$ SMs while one prefill batch runs concurrently on $S_p$ SMs. The total latency of such a configuration is expressed as:

$$t(S_p, S_d, k) = \max\big(k\,t_d(S_d),\, t_p(S_p)\big),$$

which indicates that residual compute bubbles may still occur on either the decode or prefill side, depending on the chosen configuration.

Let $T_{\text{decode}}$ denote the number of tokens produced per decode step across the scheduled decode requests, and $T_{\text{prefill}}$ denote the number of tokens in the scheduled prefill batch. The scheduler seeks the configuration $(S_p, S_d, k)$ that maximizes total token throughput under the latency constraint:

$$\max_{S_p, S_d, k} \quad \frac{k\,T_{\text{decode}} + T_{\text{prefill}}}{\max\big(k\,t_d(S_d),\, t_p(S_p)\big)}$$
$$\text{s.t.} \quad t_d(S_d) \leq \tau_{\text{TBT}},$$

where $\tau_{\text{TBT}}$ is the predefined TBT latency bound.

Note that this optimization naturally favors allocating more SMs to the prefill task to reduce its latency while assigning the minimal SMs required for decode to just satisfy the TBT constraint, since prefill contributes more substantially to total throughput. In practice, the scheduler enumerates feasible values of $S_d$, discards configurations violating the TBT constraint, and calculates the throughput of each configuration to solve the optimization problem with negligible CPU overhead.

### 4.3. Interruption-Free Kernel Dispatching and Look-Ahead Decode Execution

To enable concurrent prefill and decode execution, Duet-Serve initializes two dedicated CUDA streams, one for decode and one for prefill. After determining the optimal SM partitioning configuration, the scheduler invokes `libsmctrl` to bind each stream to its designated SM region, ensuring that the two workloads execute independently without interference. GPU kernels for both batches are then dispatched concurrently by the CPU within their respective stream contexts.

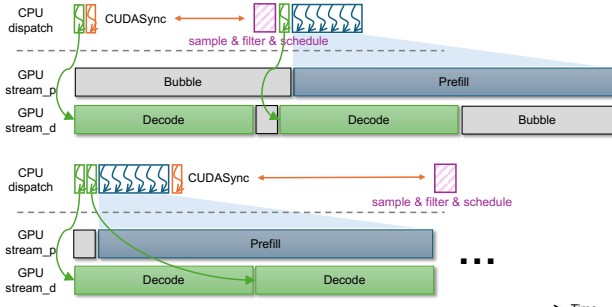

*Figure 5.* (Top) Conventional decoding incurs CPU–GPU stalls from per-step synchronization. (Bottom) Interruption-free kernel dispatching by scheduling multiple decoding steps in advance.

To minimize CPU-side dispatch overhead, DuetServe leverages CUDA Graph capture for decode execution. During initialization, the system records the sequence of decode kernels into a reusable CUDA Graph (NVIDIA, 2025b), enabling efficient graph replay with negligible launch latency. Prefill execution, however, cannot be similarly captured because its attention kernels often exhibit dynamic tensor shapes and variable control flows (vLLM, 2025a). Consequently, prefill kernels are launched individually by the CPU, while decode kernels are launched collectively through the cached CUDA Graph. Since launching a CUDA Graph introduces less than 0.5 ms of overhead, compared to tens of milliseconds for prefill kernel dispatch, the scheduler always initiates decode execution first to prevent CPU-induced stalls, as shown in Figure 5.

To further reduce synchronization overhead between consecutive decode steps, DuetServe introduces a look-ahead decode execution mechanism. In conventional decoding, each iteration involves CPU-side synchronization to sample tokens, filter completed requests, update KV cache mappings, and prepare input metadata for the next step. These operations create frequent CPU–GPU stalls that limit overlap with prefill. The look-ahead strategy eliminates these interruptions by preallocating multiple KV cache slots per request and preparing all metadata for $k$ future decode steps in advance. The CPU then launches $k$ pre-recorded CUDA Graphs consecutively without waiting for intermediate synchronization, allowing continuous GPU execution across multiple decode iterations.

## 5. Evaluation

### 5.1. Experimental Setup

**Testbed.** Experiments are conducted on a server equipped with a 96-core Intel Xeon Platinum 8480C CPU and two NVIDIA H100 GPUs (80 GB) connected via NVLink, running driver version 580.95.05 and CUDA 13.0. All experiments use PyTorch 2.8.0.

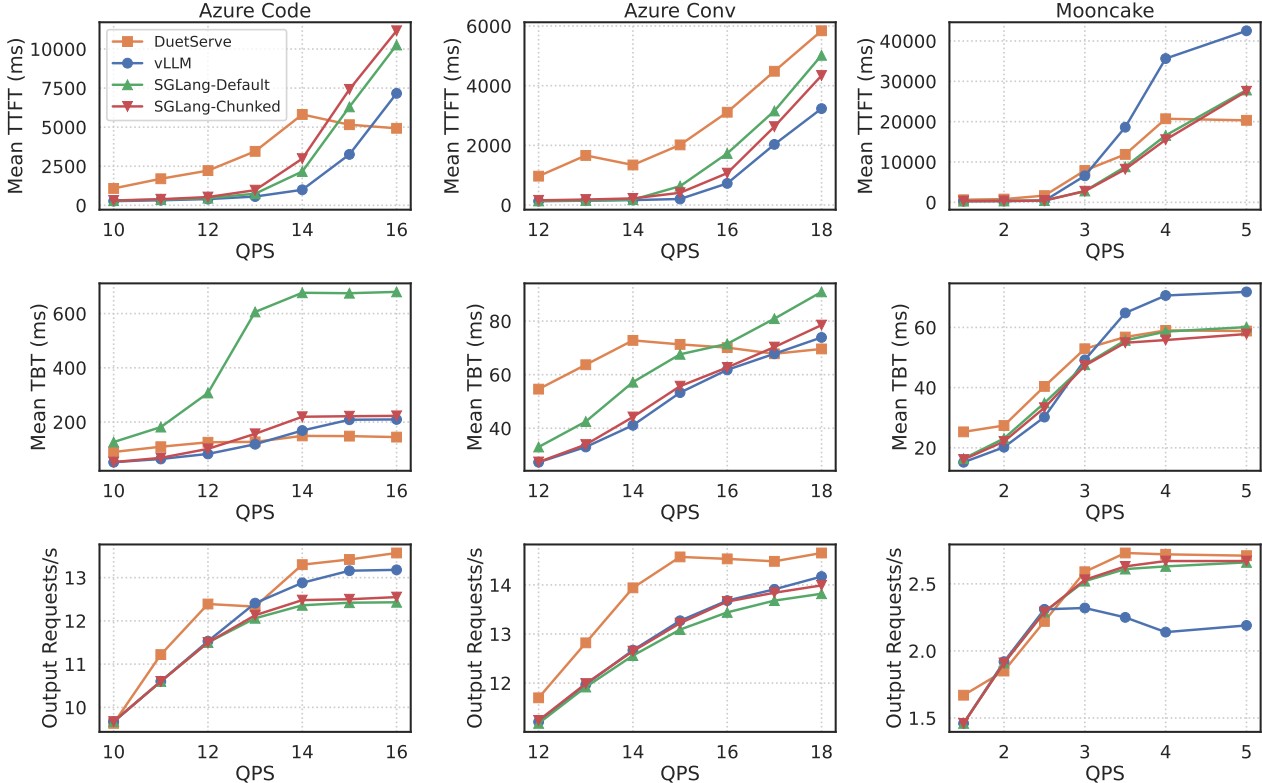

*Figure 6.* The end-to-end performance of different workloads with Qwen3-8B.

**Implementation.** We implement DuetServe from scratch with approximately 3,000 lines of Python code. The system integrates the `libsmctrl` CUDA library through a Python API to control GPU SM allocation per CUDA stream. It incorporates essential online serving optimizations, including paged KV cache management (Kwon et al., 2023), continuous batching (Yu et al., 2022), CUDA Graph capture and replay (NVIDIA, 2025b), `torch.compile` kernel fusion (PyTorch, 2025), and the FlashAttention-3 kernel (Shah et al., 2024). Tensor parallel inference is implemented using NCCL to handle inter-GPU communication.

**Models and Workloads.** We evaluate two representative LLMs: Qwen3-8B (TP = 1) and Qwen3-14B (TP = 2). Three real-world LLM serving traces are used as workloads: Azure Code (Microsoft, 2023), Azure Conversation (Microsoft, 2023), and Mooncake Conversation (Qin et al., 2025). These traces exhibit distinct usage patterns and token length distributions, summarized in Table 1. For Mooncake Conversation, we sample 1,000 requests due to the trace's large scale. Following prior work (Yu et al., 2022; Kwon et al., 2023), we model request arrivals using a Poisson process to simulate realistic serving dynamics.

**Baselines.** We compare DuetServe with four state-of-the-art LLM serving systems under both PD aggregation and

*Table 1.* Workload traces used for evaluations.

| TRACE | # REQUESTS | ISL | OSL |
|---|---|---|---|
| AZURE-CODE | 19366 | 2047 | 28 |
| AZURE-CONV | 8819 | 1155 | 211 |
| MOONCAKE | 1000 | 12035 | 343 |

PD disaggregation settings:

- `vLLM` (v0.10.1–v1): employs the default chunked prefill scheduler with a token budget of 8192 on H100.
- `SGLang-Default` (v0.5.0): employs a throughput-oriented scheduler that opportunistically executes prefill-only batches when sufficient GPU memory is available for several consecutive iterations, before switching to decode-only iterations to drain pending requests.
- `SGLang-Chunked`: configured with serving argument `enable-mixed-chunk` to enable the Sarathi-Serve chunked-prefill scheduler, using a token budget of 8192.
- `Dynamo` (v0.5.1) with `vLLM` backend: implements PD disaggregation in a 2-GPU setup, assigning one GPU to the prefill phase (1P) and the other to the decode phase (1D). KV cache transfer is managed via the NIXL library, optimized for P2P GPU communication.

All baselines employ consistent configurations for fair comparison: a maximum batch size of 1024, a GPU memory

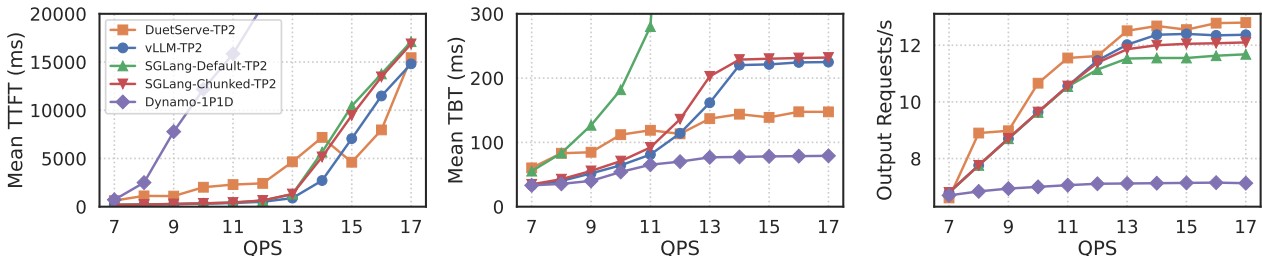

*Figure 7.* The end-to-end performance of Azure-Code workload with Qwen3-14B (TP = 2).

utilization ratio of 0.9, and FlashAttention-3 (Shah et al., 2024) as the attention backend.

**Metrics.** We report three key metrics: mean TTFT, mean TBT, and output request throughput. Throughput is defined as the total number of completed requests divided by the end-to-end serving duration, indicating the system's maximum serving capacity. An efficient serving system should achieve low latency while sustaining high throughput under increasing request loads.

### 5.2. End-to-end Performance

As shown in Figure 6, across the three workloads using Qwen3-8B (TP = 1), DuetServe consistently achieves the lowest TBT and highest output request throughput. In Azure-Code, DuetServe maintains TBT below 150 ms at QPS = 16, whereas both vLLM and SGLang-Chunked exceed 200 ms. Notably, the SGLang-Default baseline exhibits unbounded TBT growth, surpassing 680 ms, as it repeatedly inserts prefill-only batches that interrupt decode generations. By prioritizing decode progress and executing prefill concurrently, DuetServe (13.57 req/s) achieves 1.1× higher throughput than SGLang-Default (12.43 req/s) at QPS = 16. Although DuetServe's TTFT is slightly higher at light load, this reflects an intentional design trade-off: the system safeguards the decode TBT SLO while opportunistically advancing prefill using leftover GPU resources. Prefill latency, and consequently TTFT, can be relaxed under heavy workloads to favor decode responsiveness, improving steady-state throughput. Hence, modest TTFT fluctuations indicate that the GPU is fully utilized rather than inefficiently scheduled.

In Azure-Conv and Mooncake workloads, the same pattern persists. DuetServe bounds TTFT growth even beyond QPS = 15 and 3, respectively, demonstrating stronger queue stability and balanced SM utilization. More importantly, its TBT remains consistently lower than both vLLM and SGLang, achieving up to 1.3× higher output request throughput than vLLM at peak load in Mooncake. Specifically, vLLM reaches 2.14 req/s, while DuetServe sustains 2.72 req/s at QPS = 5. This improvement stems from Duet-Serve's ability to execute decode steps independently of prefill synchronization, allowing requests to complete faster. While vLLM achieves the lowest TTFT at light load, Duet-Serve's higher throughput and stable TBT under contention underscore its effectiveness in sustaining low-latency token generation under full GPU saturation.

### 5.3. Multi-GPU Inference Performance

We extend the experiments to a multi-GPU environment. In the PD aggregated setting, we adopt a tensor parallelism (TP) degree of 2, evenly splitting model weights across two GPUs. Both vLLM and SGLang baselines use the same TP configuration for fair comparison. In the PD disaggregated setting, Dynamo assigns one GPU as the prefill worker and another as the decode worker (1P+1D). This 1P+1D split is dictated by our two-GPU testbed; with more GPUs, device-level disaggregation can adopt richer PD ratios (e.g., 2P+6D) but still partitions at whole-GPU granularity, whereas Duet-Serve partitions at the finer SM granularity (Appendix B).

As shown in Figure 7, DuetServe-TP2 achieves the second-lowest TBT while maintaining the highest throughput across all QPS levels. While vLLM-TP2 and SGLang-Chunked-TP2 exhibit rising TBT beyond 200 ms once QPS exceeds 13, DuetServe sustains stable per-token latency below 150 ms even under saturation. The SGLang-Default-TP2 baseline again suffers from unbounded TBT growth due to frequent prefill interruptions, whereas Dynamo-1P1D achieves the lowest TBT but at the expense of throughput. Its prefill GPU becomes the bottleneck, leaving the decode GPU underutilized because its throughput cannot keep pace with the decode stage. Overall, these results confirm that DuetServe's adaptive spatial multiplexing scales effectively under multi-GPU execution, maintaining low decoding latency while avoiding the inefficiency and imbalance inherent in fully disaggregated PD configurations.

Appendix A includes additional ablations to validate key design choices. We evaluate roofline predictor accuracy across TPC counts, compare DuetServe with static SM partitions, and study sensitivity to different prefill–decode

lengths, showing larger gains for prefill-heavy workloads. Appendix B further discusses scaling to larger clusters, communication modeling, and system limitations.

## 6. Related Work

**LLM Serving Frameworks.** Early LLM serving systems focused on unified execution to improve throughput and memory efficiency. ORCA (Yu et al., 2022) introduced iteration-level scheduling (i.e., continuous batching), admitting and retiring requests at each step to reduce request latency and improve GPU utilization. FastServe (Wu et al., 2026) later challenged the standard first-come, first-served policy, proposing a skip-join multi-level feedback queue scheduler that preempts at token granularity to mitigate head-of-line blocking from long requests. vLLM (Kwon et al., 2023) addressed KV cache memory fragmentation via PagedAttention, by dividing memory space into blocks and allocating them to requests on demand. SGLang (Zheng et al., 2024) proposed RadixAttention to maximize KV reuse across multi-turn generations through a prefix-tree cache. Sarathi-Serve (Agrawal et al., 2024b) improved efficiency through chunked prefill, bounding per-iteration work with a token budget to reduce pipeline bubbles. DeepSpeed-FastGen (Holmes et al., 2024) adopts a similar Dynamic SplitFuse strategy that composes prefill and decode tokens into uniform-sized batches to sustain high throughput.

**GPU Resource Sharing.** GPU resource sharing focuses on efficient allocation among multiple ML workloads. GSLICE (Dhakal et al., 2020) and KRISP (Chow et al., 2023) apply spatial partitioning to colocate tasks, while Orion (Strati et al., 2024), REEF (Han et al., 2022), and BLESS (Zhang et al., 2025) leverage fine-grained temporal multiplexing for DNN inference. However, these target general ML workloads, not LLM-specific needs. MuxServe (Duan et al., 2024) addresses this gap by combining spatial and temporal sharing to colocate multiple LLMs, allocating separate GPU resources to prefill and decode. MuxWise (Chen et al., 2025) partitions SMs statically for concurrent execution of prefill and decode phases, whereas Bullet (Lin et al., 2026) and Semi-PD (Hong et al., 2025) dynamically adapt SM allocation using feedback loops and latency models. Nexus (Shi et al., 2025) further optimizes GPU spatial sharing with a runtime KV-aware analytical model. At the kernel level, NanoFlow (Zhu et al., 2025) overlaps compute, memory, and network operations, while POD-Attention (Kamath et al., 2025) fuses prefill and decode attention kernels into a single one to maximize SM utilization.

**PD Disaggregated Systems.** To reduce phase interference, recent systems decouple prefill and decode across heterogeneous hardware. DistServe (Zhong et al., 2024) assigns each phase to separate GPUs and searches for op-timal model-parallel strategies per phase. Splitwise (Patel et al., 2024) explores both homogeneous and heterogeneous device configurations to optimize cost, throughput, and power efficiency. TetriInfer (Hu et al., 2024b) employs a two-level scheduling algorithm with resource prediction to balance load across GPUs. Mooncake (Qin et al., 2025) and MemServe (Hu et al., 2024a) introduce distributed KV cache memory pools to enable cache reuse, while Wind-Serve (Feng et al., 2025) reduces GPU underutilization through stream-based dynamic rescheduling across prefill and decode instances. Dynamo (NVIDIA, 2025c) further proposes KV-aware request routing and multi-level memory KV cache offloading.

However, these approaches add system complexity through cross-instance orchestration and KV cache transfer overhead, and adapt poorly to dynamic traffic: the prefill-to-decode ratio is fixed at coarse, whole-GPU granularity, so rebalancing it demands costly reconfiguration that is too slow to track request-level fluctuations (Appendix B). DuetServe instead attains disaggregation-level isolation within a single device at fine SM granularity, avoiding both inter-GPU KV transfers and runtime reconfiguration overhead.

## 7. Conclusion

We presented DuetServe, an adaptive framework that unifies the high utilization of aggregated serving with the isolation of disaggregation. Guided by an attention-aware roofline model, DuetServe dynamically partitions SMs to decouple prefill and decode on a single GPU, while its look-ahead dispatch engine minimizes synchronization overheads. Our evaluations confirm that DuetServe maximizes hardware efficiency, achieving up to $1.3\times$ higher throughput than state-of-the-art baselines while maintaining low Time-Between-Tokens service level objectives.

## Impact Statement

This paper presents system research aimed at improving the efficiency and reliability of real-time LLM serving. While such advances can enable broader deployment and lower operational cost, we do not anticipate societal consequences that require specific emphasis beyond standard considerations for responsible use of LLM-based systems.

## Acknowledgment

We sincerely thank all the reviewers for their time and constructive comments. This material is based upon work supported by NSF award number 2224319, REAL@USC-Meta center, and VMware gift. We thank Yongqin Wang for his helpful discussions and feedback during this work.

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

# A. Ablation Study

**Roofline Modeling Accuracy.** Figure 8 compares the profiled latency with the latency estimated by the roofline analytical model for both the $8 \times 1024$ prefill and $16 \times 1024$ decode workloads on Qwen3-8B (TP = 1) and Qwen3-14B (TP = 2). For the compute-bound prefill phase, the model tracks the measured latency closely across different TPC counts, showing nearly linear latency reduction until about 40 TPCs, after which the curve flattens as compute throughput saturates. This confirms that prefill latency is dominated by SM compute utilization rather than memory bandwidth.

For decode kernels, especially when the decode stream is assigned only a small number of TPCs, the model is intentionally conservative and tends to overestimate latency. This can introduce some compute bubbles on the decode side. However, this typically does not change the optimal partition by much: the objective is to maximize overall token throughput, and decode contributes relatively little to throughput because each decode request produces only one token per step and decode batches generate far fewer tokens than prefill batches.

In contrast, underestimating decode latency is more harmful. If decode takes longer than predicted, the optimizer may allocate too few resources to decode and over-provision prefill. This can force prefill to wait at synchronization points, creating bubbles on the prefill side and reducing overall token throughput. We can calibrate the roofline model to better match decode latency; however, in our experiments this calibration does not lead to a noticeable performance improvement.

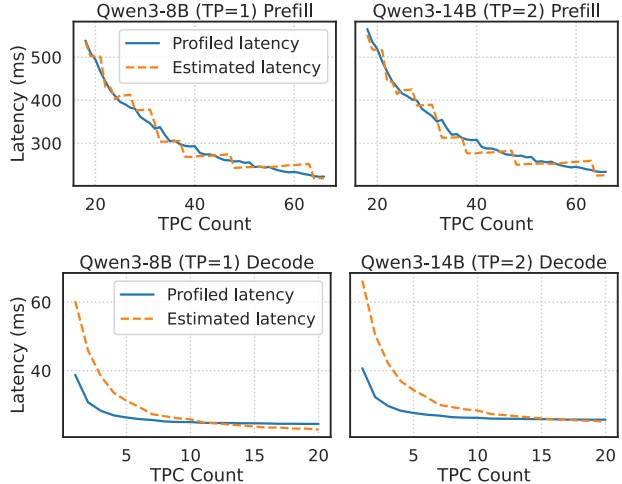

*Figure 8.* Profiled and roofline-modeled latency comparison.

**Static SM Partitioning.** Figure 9 compares system throughput under different static SM partitioning ratios for Qwen3-8B (TP = 1) and Qwen3-14B (TP = 2) across the Azure-Code, Azure-Conv, and Mooncake workloads. The three configurations `Sd22-Sp44`, `Sd33-Sp33`, and `Sd44-Sp22` represent fixed SM allocations between the decode and prefill phases. Throughput varies noticeably across workloads, confirming that static partitioning leads to persistent imbalance: either idle compute resources in one phase or congestion in the other. In contrast, DuetServe's adaptive scheduling dynamically reallocates SMs at runtime, maintaining balanced utilization and higher concurrency under diverse traffic conditions.

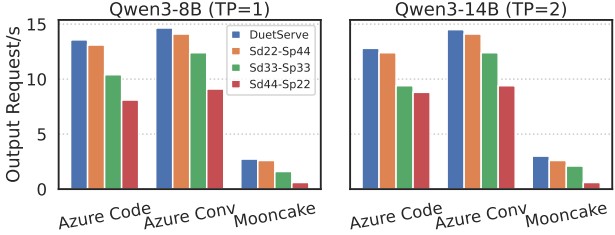

*Figure 9.* Throughput comparison across workloads and models using static SM partitioning versus DuetServe.

**Workload Sensitivity.** We evaluate DuetServe on synthetic workloads by fixing the input sequence length (ISL) and varying the output sequence length (OSL). Table 2 compares DuetServe with `vLLM` across three workloads under maximum serving capacity. When the generation is short, DuetServe provides the largest benefit. As the workload becomes more decode-dominant, the TBT improvements become marginal. This trend is consistent with observations in recent PD disaggregation studies (Mitra et al., 2026; NVIDIA, 2025a): isolating prefill from decode is most beneficial when prefill contributes a large fraction of iteration latency, while decode-heavy regimes inherently exhibit less prefill–decode contention and thus approach the behavior of PD aggregation.

DuetServe operates in PD aggregated mode by default and activates PD-style isolation via SM partitioning only when the roofline predictor flags a potential TBT violation. Consequently, in decode-heavy regimes where interference is limited, DuetServe largely remains in aggregated execution and avoids the persistent costs of conventional PD disaggregation, such as duplicating model weights and transferring KV caches between prefill and decode resources.

*Table 2.* Performance comparison between DuetServe and vLLM under synthetic workloads.

| ISL | OSL | ISL/OSL | THROUGHPUT (REQ/S) | MEAN TBT (MS) | THROUGHPUT GAIN |
| --- | --- | --- | --- | --- | --- |
| | | | vLLM → DUETSERVE | vLLM → DUETSERVE | (DUETSERVE / vLLM) |
| 4096 | 64 | 64 | $10.0 \to 12.8$ | $170 \to 105$ | $1.28\times$ |
| 4096 | 1024 | 4 | $8.8 \to 9.8$ | $55 \to 50$ | $1.11\times$ |
| 4096 | 2048 | 2 | $6.9 \to 7.2$ | $45 \to 44$ | $1.04\times$ |

**Latency Breakdown.** Figure 10 shows CPU and GPU activities and SM utilization over two consecutive iterations during LLM serving. In the first iteration, the scheduler assigns 48 TPCs to the prefill stream and 18 TPCs to the decode stream, running five decode steps before synchronizing with prefill execution. The CPU scheduling overhead, including solving the optimal GPU partitioning configuration, remains below 1 ms. Decode kernels are launched first, causing a short delay before prefill execution begins, but both streams sustain high overlap and stable SM efficiency. In the second iteration, the execution switches back to PD aggregated mode on the main stream, where prefill and decode requests execute synchronously in a unified context. This transition highlights DuetServe's adaptive scheduler, which alternates between spatial and temporal sharing based on workload characteristics and predicted latency.

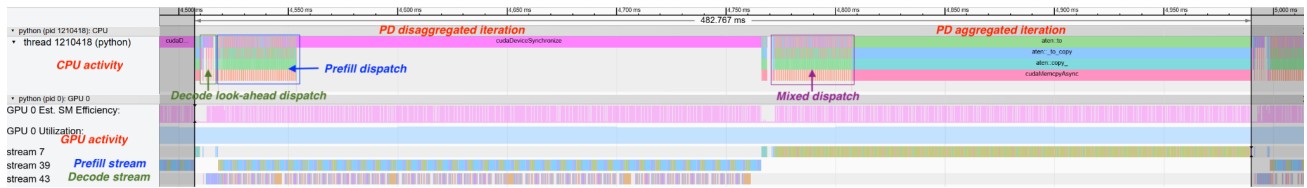

*Figure 10.* Profiled CPU and GPU activities showing DuetServe's concurrent prefill and decode execution.

## B. Discussion and Limitations

**Cluster Size and PD-Ratio Flexibility.** With a larger GPU pool, device-level PD disaggregation gains a richer partition space (e.g., 2P+6D on eight GPUs) that can better match a workload. DuetServe is orthogonal: it partitions at SM granularity (66 TPC groups on H100), enabling finer allocation that remains useful in large deployments. To illustrate this, Table 3 compares DuetServe (TP = 8) against `Dynamo` on eight H100 GPUs for Qwen3-32B on Azure-Conv at QPS = 24. `Dynamo` starts at 4P+4D and lets its planner reconfigure to the best ratio at runtime, but its throughput suffers from the cost of PD reconfiguration: switching a GPU's role preempts in-flight decode requests, and the new prefill GPU must reload and warm up the model and rebuild its KV cache, taking roughly 40 s, during which the system runs at reduced capacity. DuetServe resolves allocation at each scheduling iteration with negligible reconfiguration latency and no in-flight request loss, achieving $1.4\times$ higher throughput, lower TTFT, and 93.5% average GPU utilization; `Dynamo`'s lower TBT reflects underutilized decode workers, consistent with its 74.6% utilization. Due to resource limits, this experiment is confined to a single 8-GPU node. More broadly, device-level disaggregation faces challenges DuetServe avoids: (i) the optimal PD ratio depends on the output-length distribution, which is generally unknown at serving time, so a mismatched ratio underutilizes resources until reconfiguration; (ii) switching a GPU's role requires reallocating KV-cache memory and reloading the model, far too slow

for request-level fluctuations; and (iii) duplicated weights and cross-GPU KV-cache transfers raise memory and bandwidth pressure. Characterizing how DuetServe's advantage scales with cluster size and PD-ratio flexibility is future work.

*Table 3.* Eight-GPU comparison on Azure-Conv (Qwen3-32B, QPS = 24). `Dynamo` starts at 4P+4D and lets its planner reconfigure (e.g., to 6P+2D or 2P+6D); DuetServe runs TP = 8.

| SYSTEM | THROUGHPUT (REQ/S) | TTFT (S) | TBT (MS) | AVG GPU UTIL. |
|---|---|---|---|---|
| DYNAMO | 5.69 | 110.2 | 23.1 | 74.6% |
| DUETSERVE | 8.02 | 58.9 | 104.7 | 93.5% |

**Communication Modeling.**  Our roofline model captures intra-node NVLink communication for tensor parallelism (Section 4.1). Extending it to multi-node clusters with diverse network topologies would require richer communication models; analytical and learned frameworks such as ASTRA-sim (Won et al., 2023) and MimicNet (Zhang et al., 2021) are promising components to integrate for large-scale prediction.

**Failure Handling and Heterogeneity.**  The roofline model predicts prefill latency accurately and is intentionally conservative for decode under small TPC allocations (Appendix A). In our evaluation, SLO misses were rare and confined to extreme saturation, where the arrival rate exceeds peak capacity and no scheduling policy can fully prevent violations. Heterogeneous deployments (e.g., compute-optimized GPUs for prefill, memory-optimized GPUs for decode) are a further direction, though current datacenters typically co-locate high compute and high memory capacity on the same server-grade GPUs and favor homogeneous clusters for operational simplicity.

