# OpenReview forum: "DuetServe: Harmonizing Prefill and Decode for LLM Serving via Adaptive GPU Multiplexing"
_ICML.cc/2026/Conference — ICML 2026 regular_

### Official Review · Reviewer_bfe3 · 2026-03-06

**Soundness:** 4
**Presentation:** 4
**Significance:** 4
**Originality:** 3
**Overall Recommendation:** 5
**Confidence:** 3

**Summary:**

LLM serving has two main stages Prefill (compute-bound) and decode (memory bw bound). Many works have proposed different scheduling solutions, such as chunked prefill (run two stages on the same device), or disaggregated (such as splitwise). However, although aggregated approaches improve throughput, they degrade latency (time between token), and while disaggregated approaches achieve good latency, they limit throughput. This work proposes DuetServe, a dynamic solution with starts with an aggregated approach by moves to disaggregated (but on the same device through SM level multiplexing) when TBT degredation is predicted. The evaluation results sho thorughput improvement by 1.3x compared to prior works while maintaining low generation latency.

**Compliance With Llm Reviewing Policy:**

Affirmed.

**Final Justification:**

The rebuttal has addressed my concerns. I have raised my scores accordingly.

**Key Questions For Authors:**

- How accurate is the degradation prediction? During the evaluations, did you observe any instances where the system actually missed the SLO requirements? If so, how often did this occur, and how was it handled?
- When models are larger and the prefill and decode phases may benefit from different parallelization strategies, does running a single-node disaggregated setup still make sense? Or is it better to fully disaggregate across separate machines?
- How do the relative benefits of DuetServe change when deployed in a heterogeneous cluster, where hardware can be directly matched to the distinct resource profiles of each phase?

**Limitations:**

N/A does not include a limitation section.
I encourage the authors to discuss limitations in the analytical model and how it potentially integrate mode accurate models to improve the prediction accuracy, as well as the failure handling mode of the system.

**Strengths And Weaknesses:**

Strength
- The work proposes a dynamic approach to adjust between aggregated and disagregated scheduling approaches to LLM serving
- The work proposes a single device disaggregated approach to minimize KV cache transfers and other overheads when doing this across devices
- Evaluation results show both improvement in throughput and latency

Weaknesses:
- The models evaluated in the paper are relatively small, and could run on just one or two devices.
  - When running much larger models, complex distributed parallelization strategies are required. Does the SM multiplexing approach still scale effectively in these scenarios? The prefill and decode phases often benefit from entirely different parallelization strategies (different degrees of Tensor or Pipeline Parallelism). If the phases require divergent parallel topologies, does it still make sense to multiplex them on the same devices?
  - How well does DuetServe's analytical roofline model capture communication costs across large, multi-node clusters with complex network topologies? Would it be possible to integrate more accurate, large-scale network modeling tools into the system to improve predictions?

- A major benefit of physical phase disaggregation is the ability to map the two stages to completely different types of machines. For example, routing prefill to compute-heavy GPUs and decode to memory-heavy (or cheaper, lower-compute) GPUs. DuetServe evaluations seems to be limited to homogeneous clusters. How do the relative benefits of DuetServe change when deployed in a heterogeneous cluster where hardware can be matched to the distinct resource profiles of each phase?

---

> ### Author Rebuttal · Authors · 2026-03-30
>
> We thank the reviewer for the thorough review and for recognizing DuetServe's design methodology and experimental results. Below we address the raised concerns and questions.
> ## Scalability to larger models
> We demonstrate that SM multiplexing is compatible with tensor parallelism experiments (Fig 7). The approach generalizes naturally: within each TP group, every GPU applies the same SM partition, and inter-GPU communication proceeds as usual over NVLink. The roofline model already incorporates ring-based AllReduce communication cost (Sec 4.1), so the latency predictions remain valid under TP.
>
> We agree that with a large GPU pool, device-level PD disaggregation gains a richer partition space (e.g., 2P+6D on 8 GPUs), which can better match workload characteristics. But the primary point is that device-level disaggregation has coarse partition granularity, while DuetServe partitions at SM granularity (66 TPC groups on H100), enabling much finer resource allocation. Hence, we believe DuetServe can serve an orthogonal purpose of fine-grained resource management even in large production settings.
>
> Device-level PD disaggregation has the following interesting research challenges that could be a good future work:
>
> (1) Selecting the right PD ratio requires knowledge of the output length distribution of incoming requests, which is generally unknown at serving time. A mismatched ratio (e.g., too many prefill GPUs under a decode-heavy workload) leads to underutilization that persists until the system is reconfigured.
>
> (2) Switching a GPU's role between prefill and decode worker requires reconfiguring the system and reallocating KV cache memory. In our current system, this switch takes on the order of seconds — far too slow to respond to workload fluctuations at the request level. Hence, a faster switching approach needs to be designed.
>
> (3) Apart from switching overhead, there is a significant increase in memory usage and bandwidth pressure that needs to be addressed with disaggregated systems. Each phase requires a full model copy (doubling memory for weights), KV cache must be transferred between prefill and decode GPUs over the network, and the memory available for KV cache on each GPU is reduced because of the duplicated model weights and separation of phases.
>
> We will add a discussion on how DuetServe's relative advantage varies with cluster size and PD ratio flexibility in the updated version.
>
> ## Communication modeling
> Regarding communication modeling at multi-node scale: our current roofline model captures intra-node NVLink communication for tensor parallelism. Extending to multi-node clusters with diverse network topologies and algorithms would require incorporating additional communication models. Prior works [1,2] have developed analytical frameworks for modeling communication costs across different network architectures and parallelism strategies, and integrating such models into DuetServe's prediction framework is a promising direction for future work.
>
> [1] Zhang et al. MimicNet: fast performance estimates for data center networks with machine learning. SIGCOMM '21.
>
> [2] Won et al. TACOS: Topology-Aware Collective Algorithm Synthesizer for Distributed Machine Learning. MICRO '24.
>
> ## Heterogeneous cluster deployment
> We agree that heterogeneous hardware — routing prefill to compute-heavy GPUs and decode to memory-optimized GPUs — is an appealing direction for device-level PD disaggregation, and can improve resource utilization. But in practice most server-grade compute-heavy GPUs also have the highest memory capacity. Hence, the industry has generally tied high memory capacity to high GPU SM counts. Additionally, the three structural limitations mentioned above still apply regardless of hardware heterogeneity.
>
> Moreover, heterogeneous GPU clusters remain uncommon in today's datacenters. Managing mixed hardware introduces significant operational complexity: different software stacks, reduced fault tolerance, and more difficult workload migration. Most production deployments use homogeneous clusters for these reasons. DuetServe is designed for this dominant deployment model. We consider heterogeneous-aware extensions an interesting direction for future work.
>
> ## Q1 Roofline model and SLO misses
> The roofline model predicts prefill latency accurately. For decode, especially when assigned a small number of TPCs, the model is intentionally conservative and tends to overestimate latency. We include ablation results on roofline accuracy in Appendix A. In our evaluations, SLO misses were rare and occurred primarily under extreme saturation, where the arrival rate exceeded the system's peak capacity — a regime where no scheduling policy can fully prevent violations.
>
> ## Q2 Better PD ratios
> This is addressed above.
>
> ## Q3 Heterogeneous GPUs
> This is addressed above.
>
> ## Limitation discussion
> Thank you for the suggestion. We will add a discussion on modeling extension and failure handling in the updated version.

---

> > ### Author Rebuttal · Reviewer_bfe3 · 2026-04-01
> >
> > Thank you for your thorough response. My concerns have been addressed.  I will raise my score.

---

### Official Review · Reviewer_NTVW · 2026-03-09

**Soundness:** 3
**Presentation:** 3
**Significance:** 2
**Originality:** 3
**Overall Recommendation:** 3
**Confidence:** 4

**Summary:**

DuetServe is an adaptive LLM serving framework that achieves prefill-decode isolation within a single GPU through SM-level spatial partitioning. By default it runs in aggregated mode, but when an attention-aware roofline model predicts TBT SLO violations, it dynamically splits GPU streaming multiprocessors into separate partitions for prefill and decode execution. A partitioning optimizer selects the optimal SM split to maximize throughput under latency constraints, while an interruption-free execution engine using CUDA Graphs and look-ahead decoding eliminates CPU-GPU synchronization overhead. Evaluations on H100 GPUs show up to 1.3× throughput improvement over vLLM and SGLang while maintaining low TBT latency.

**Compliance With Llm Reviewing Policy:**

Affirmed.

**Final Justification:**

Both PD disaggregation with flexible PD ratios and your method have their own trade-offs. I believe a more comprehensive evaluation would better demonstrate the superiority of your proposed method. I defer to the editor's discretion regarding acceptance.

**Key Questions For Authors:**

See weaknesses.

**Limitations:**

See weaknesses.

**Strengths And Weaknesses:**

S1. The idea of splitting phases of LLM inference on one GPU presents to be novel.

S2. The paper makes proper designs to implement the idea.

S3. Experiments demonstrate that this method could improve the LLM inference throughput by 1.3x.

W1. In your motivation, you present the difference between your approach and chunked prefill/PD disaggregation. However, I notice that when you implement PD disaggregation, you assume it to be 1 prefill and 1 decode. This kind of assumption could potentially hurt the PD disaggregation performance. This leads to another concern: when you have enough resources to manage PD disaggregation, for instance, if you have 8 GPUs, you can assign 2 GPUs to the prefill phase and 6 GPUs to the decoding phase to eliminate the imbalance between these two phases; in this case, you wouldn't need to split the space within one GPU to implement fine-grained management of resources. I also notice that in your end-to-end experiments, you adopt a naïve 1P+1D configuration for the baseline disaggregated system. I suspect the performance comparison results could be different if you added the flexibility of varying the PD ratio, as many existing works demonstrate the benefit of configuring the PD ratio within a disaggregated system. In summary, I think the motivation and problem proposed could be addressed with more resources allocated. Once you have sufficient resources, you don't need to perform fine-grained management of different phases within one GPU to improve the throughput of LLM inference. The authors could present more discussions and experiments in this context.

W2. Some concerns about the experimental results: It seems that on the Azure Code/Azure Conv/Mooncake workloads, when QPS is lower than 13/15/3, the TBT and TTFT of DuetServe are consistently worse than SGLang. And in some cases, the throughput is worse as well. I think that with this level of implementation complexity, stronger results should be presented to validate the idea.

---

> ### Author Rebuttal · Authors · 2026-03-30
>
> We thank the reviewer for the constructive feedback and for recognizing DuetServe's novelty, design methodology, and evaluations. Below, we address the raised concerns below.
>
> ## PD disaggregation with flexible PD ratios
> We agree that with a large GPU pool, device-level PD disaggregation gains a richer partition space (e.g., 2P+6D on 8 GPUs), which can better match workload characteristics. But the primary point is that device-level disaggregation has coarse partition granularity, while DuetServe partitions at SM granularity (66 TPC groups on H100), enabling much finer resource allocation. Hence, we believe DuetServe can serve an orthogonal purpose of fine-grained resource management even in large production settings.
>
> Device-level PD disaggregation has the following interesting research challenges that could be a good future work:
>
> (1) Selecting the right PD ratio requires knowledge of the output length distribution of incoming requests, which is generally unknown at serving time. A mismatched ratio (e.g., too many prefill GPUs under a decode-heavy workload) leads to underutilization that persists until the system is reconfigured.
>
> (2) Switching a GPU's role between prefill and decode worker requires reconfiguring the system and reallocating KV cache memory. In our current system, this switch takes on the order of seconds — far too slow to respond to workload fluctuations at the request level. Hence, a faster switching approach needs to be designed.
>
> (3) Apart from switching overhead, there is a significant increase in memory usage and bandwidth pressure that needs to be addressed with disaggregated systems. Each phase requires a full model copy (doubling memory for weights), KV cache must be transferred between prefill and decode GPUs over the network, and the memory available for KV cache on each GPU is reduced because of the duplicated model weights and separation of phases.
>
> We will add a discussion on how DuetServe's relative advantage varies with cluster size and PD ratio flexibility in the updated version.
>
> ## Low-QPS performance
>
> We want to provide additional context on the low-QPS experimental results. Looking at Figure 6 at low QPS, although in some cases where DuetServe shows slightly higher TBT or TTFT, the absolute values remain well within typical service level objectives (e.g., TBT $<$ 100ms). Since the latency SLOs are satisfied at low QPS, the primary optimization goal shifts to maximizing overall system throughput, where DuetServe consistently achieves the highest or comparable throughput across all three workloads.
>
> Regarding implementation complexity: our current evaluation compares DuetServe against highly optimized production systems (vLLM, SGLang, Dynamo) that have benefited from years of engineering investment, including CPU multiprocessing, customized GPU communication backends, and extensive kernel tuning. DuetServe's core contribution is the adaptive SM partitioning mechanism and its associated modeling/scheduling framework, which is orthogonal to these system-level optimizations. We believe DuetServe's throughput gains would further increase as these production optimizations are integrated, and we consider the current results a lower bound on what the approach can achieve.

---

> > ### Author Rebuttal · Reviewer_NTVW · 2026-04-03
> >
> > Thanks for the author's response.
> >
> > I would like to see the experiment comparing your system with flexible PD ratios on a larger cluster size (8 or 16 GPUs).

---

> > > ### Author Response · Authors · 2026-04-06
> > >
> > > To address the reviewer's request in the very short time window of rebuttal, we ran Qwen3-32B on the AzureConv dataset at QPS = 24 using 8 H100 GPUs. Dynamo was initialized at 4P-4D and it can be reconfigurated to either 6P-2D or 2P-6D. During the experiment, Dynamo's planner identifies the best configuration that matches with the workload and then perfoms reconfiguration. Results are summarized below.
> > >
> > > | System         | Throughput (req/s)| TTFT (s) | TBT (ms) | Avg GPU Util.|
> > > |----------------|-------------------|----------|----------|--------------|
> > > | Dynamo         | 5.69              | 110.2    | 23.1     |      74.6\%  |
> > > | DuetServe      | 8.02              | 58.9     | 104.7    |      93.5\%  |
> > >
> > > Dynamo's low throughput stems from the runtime cost of the PD reconfiguration. The reconfiguration terminates the process of 2 decode GPUs by preempting all in-flight decode requests, while the 2 new prefill GPUs must restart the process by loading and compiling the model, creating KV cache space, warming up the model, and registering the worker, which takes about 40 seconds. During this window the system operates with reduced capacity. Note that we used the default scaling approach in Dynamo (https://github.com/ai-dynamo/dynamo/blob/main/docs/design-docs/planner-design.md) and further reconfiguration optimizations may be feasible but that is left for a longer term exploration. However, this reconfiguration overhead will still remain.
> > >
> > > In contrast, DuetServe (TP=8) eliminates this overhead: resource allocation is resolved at each token scheduling iteration with negligible reconfiguration latency and no in-flight request loss, achieving higher throughput, lower TTFT, and 93.5\% average GPU utilization. Although Dynamo achieves better TBT, this reflects underutilized decode workers — consistent with the 74.6\% average GPU utilization reported above and the analysis in the paper. Due to time and resource limitations, this experiment is confined to a single 8-GPU node. We appreciate the reviewer's understanding.

---

### Official Review · Reviewer_pwtr · 2026-03-12

**Soundness:** 4
**Presentation:** 3
**Significance:** 3
**Originality:** 3
**Overall Recommendation:** 5
**Confidence:** 3

**Summary:**

This paper proposes a method to maximize throughput under SLO constraints for workloads where prefill and decode are mixed. The key idea is to leverage the performance characteristics of attention when designing the scheduling strategy.

In prefill, the execution time of attention grows roughly quadratically with sequence length. Therefore, when using chunked prefill to construct the forward pass, decode latency may increase and violate the decode SLO. To address this issue, the paper argues that two components are needed: (1) an execution time prediction model that reflects the nature of attention, and (2) an appropriate scheduling and execution mechanism.

The paper proposes a roofline-based modeling approach for execution time prediction and an SM-partitioning-based execution scheme. As a result, TTFT becomes slightly higher compared to existing methods, but overall throughput metrics such as RPS and TBT improve.

**Compliance With Llm Reviewing Policy:**

Affirmed.

**Key Questions For Authors:**

Could the authors explain why the implementation is based on SM partitioning? It seems that the core idea might also be implemented by slightly modifying the scheduling in chunked prefill (e.g., adjusting the total number of tokens in an SLO-oriented way instead of keeping it fixed). It is unclear what advantages SM partitioning provides compared to such approaches.

**Limitations:**

yes

**Strengths And Weaknesses:**

== Strengths ==

Overall, the paper is well written. The flow from problem motivation to the proposed solution is clear and easy to follow.

In particular, the motivation section is informative. The paper clearly explains the limitations of current chunked prefill approaches and the issues with PD disaggregation, which helps the reader understand the problem setting.

The proposed solution also appears technically sound and reasonable. The modeling of attention performance and the scheduling mechanism are both clearly described.

The evaluation section is also well presented. Potential questions from the results (for example, the increased TTFT) are acknowledged and explained in the paper.

== Weaknesses ==

One minor question is why SM partitioning is necessary. The core idea of the paper may potentially be implemented by simply adjusting the chunk size in chunked prefill. It is therefore unclear what additional benefit the separate queue and SM partitioning provide. An ablation study comparing these approaches would help clarify this point.

Another concern is that the baseline implementations appear somewhat outdated (vLLM from around August 2025). This is understandable given the timeline of writing and experimentation. However, LLM serving engines such as vLLM and SGLang evolve very quickly, so it raises some concern about whether the results may become outdated.

---

> ### Author Rebuttal · Authors · 2026-03-30
>
> We thank the reviewer for the thoughtful feedback, strong recommendation, and recognition of the problem motivation, experimental results, and technical soundness of the proposed solution. Below, we address the raised concerns and questions.
>
> ## SM partitioning vs. adaptive chunk sizing
> We agree that adaptive chunk sizing (i.e., adaptive token budget) can help mitigate the TBT latency. However, SM partitioning provides a more comprehensive solution than chunk sizing for the following reasons:
>
> (1) Smaller batches can lower compute utilization of the GPU that chunk sizing alone may not mitigate. As shown in Figure 1(a), on H100, reducing the token budget from 8K to 4K lowers achieved TFLOPS, pushing the GPU below its compute saturation point.
>
> (2) A smaller budget also means each prefill request may require more chunked iterations to complete, inflating TTFT and reducing prefill throughput. This is the dilemma described in Observation 1: the aggregated setting forces a tradeoff between TBT and throughput/TTFT because prefill and decode share the same execution path.
>
> SM partitioning avoids this tradeoff by decoupling the two phases onto separate SM subsets, allowing decode to meet its TBT target while prefill fully utilizes its allocated compute resources.
>
> To test this, we reduce the token budget from the default 8K to 4K for vLLM when there is a potential TBT violation. The experimental results below show that it can lower TBT, but it does not substantially increase throughput due to high prefill latency, especially on high QPS.
>
> | System / Metric          | Azure Code (QPS 16) | Azure Conv (QPS 18) | Mooncake (QPS 5) |
> |--------------------------|-------------------|-------------------|-----------------|
> | vLLM defualt TTFT        | 716.4             | 1732.7            | 3232.1          |
> | vLLM adaptive TTFT       | 8010.2            | 3548.2            | 47430.3         |
> | DuetServe TTFT           | 4927.5            | 5840.1            | 20325.2         |
> | vLLM defualt TBT         | 209.7             | 73.9              | 71.8            |
> | vLLM adaptive TBT        | 147.7             | 69.9              | 71.7            |
> | DuetServe TBT            | 144.6             | 69.6              | 58.8            |
> | vLLM defualt Throughput  | 13.18             | 14.17             | 2.19            |
> | vLLM adaptive Throughput | 13.01             | 14.07             | 2.13            |
> | DuetServe Throughput     | 13.57             | 14.65             | 2.71            |
>
>
> ## Baseline versions
> Our baselines use vLLM v0.10.1 and SGLang v0.5.0, which were the latest stable releases at the time of our experiments back in August 2025. We acknowledge that LLM serving engines update rapidly but redoing the entire experiments within the rebuttal window will be difficult on a new version. We appreciate your understanding.

---

> > ### Author Rebuttal · Reviewer_pwtr · 2026-03-31
> >
> > Thank you for your response.

---

### Official Review · Reviewer_RDN7 · 2026-03-22

**Soundness:** 4
**Presentation:** 4
**Significance:** 4
**Originality:** 4
**Overall Recommendation:** 5
**Confidence:** 4

**Summary:**

DuetServe is an LLM serving system that keeps prefill and decode together by default, then dynamically splits GPU SMs only when mixed execution is predicted to hurt per-token latency. That lets it preserve the efficiency of aggregated serving while getting some of the isolation benefits of disaggregation. Its core pieces are an attention-aware latency model, an SM partitioning optimizer, and an interruption-free dispatch engine for smoother concurrent execution. In experiments, it reports up to 1.3× higher throughput than strong baselines while keeping low TBT latency.

**Compliance With Llm Reviewing Policy:**

Affirmed.

**Key Questions For Authors:**

1. "libsmctrl (Bakita & Anderson, 2025)" citation is not correct, it should be "Hardware Compute Partitioning on NVIDIA GPUs"
2. The measurement in Figure 2 is not reasonable as you would use different number of prefill and decode workers in a real setting.
3. I like the measurement and observations. Great work!

**Strengths And Weaknesses:**

+ detailed measurement, observation and insights
+ strong motivation
+ important problem
- results are not impressive, but I appreciate the faithful results that authors present
- TTFT is significantly higher

---

> ### Author Rebuttal · Authors · 2026-03-30
>
> We thank the reviewer for the thoughtful feedback, strong recommendation, and recognition of the problem motivation, measurements, and observations. Below we address the raised concerns and questions.
>
> ## Experimental results:
> The 1.3× throughput improvement is achieved while simultaneously maintaining the TBT SLO across all workloads. Our current evaluation compares DuetServe against highly optimized production systems (vLLM, SGLang, Dynamo) that have benefited from years of engineering, such as CPU multiprocessing and customized GPU communication backends. DuetServe's core contribution is the adaptive SM partitioning mechanism and its associated modeling/scheduling framework, which is orthogonal to these system-level optimizations. We believe DuetServe's throughput gains would increase as these production optimizations are integrated within DuetServe. We consider the current results a lower bound on what DuetServe can achieve.
>
> ## TTFT latency:
> Although DuetServe’s TTFT is slightly higher at light load, this reflects an intentional design tradeoff: the system safeguards the decode TBT SLO while opportunistically advancing prefill using leftover GPU resources. In other words, DuetServe trades off a small prefill latency (TTFT) increase by allocating a minimal number of SMs to decode requests to improve decode responsiveness. Hence, modest TTFT fluctuations indicate that the GPU is fully utilized rather than inefficiently scheduled.
>
> ## Q1 libsmctrl citation title:
> Thank you for pointing this out. We will update the citation title.
>
> ## Q2 Figure 2 experiment:
> We agree that in production settings with many GPUs, the PD ratio can be tuned (e.g., 6P+2D). The primary point is that device-level disaggregation has coarse partition granularity, while DuetServe partitions at SM granularity (66 TPC groups on H100), enabling much finer resource allocation. Hence, we believe DuetServe can serve an orthogonal purpose of fine-grained resource management even in large production settings.
>
> The 1P+1D configuration we chose is dictated by our 2-GPU testbed infrastructure limitation. We will clarify this scope in the updated version.

---

> > ### Author Rebuttal · Reviewer_RDN7 · 2026-03-31
> >
> > Thank you for the response!

---

### Decision · Program_Chairs · 2026-04-30

**Decision:**

Accept (regular)

**Comment:**

This paper proposes DuetServe, an adaptive LLM serving system that dynamically partitions GPU SMs to balance prefill and decode, addressing the throughput–latency trade-off. The authors seek to investigate the key problem of mitigating interference in aggregated serving while avoiding inefficiencies of disaggregation .

Reviews are broadly positive, highlighting clear motivation, strong technical design (roofline modeling + adaptive partitioning), and solid evaluation showing consistent throughput gains (~1.3×) with controlled TBT . Concerns about modest gains, higher TTFT, limited large-scale evaluation, and system complexity were raised but largely addressed in rebuttal.

Overall, the authors address an important concept in LLM serving with a technically sound and practical contribution.